# Prenatal Diagnosis of Fetal Heart Failure

**DOI:** 10.3390/diagnostics13040779

**Published:** 2023-02-18

**Authors:** Kasemsri Srisupundit, Suchaya Luewan, Theera Tongsong

**Affiliations:** Department of Obstetrics and Gynecology, Faculty of Medicine, Chiang Mai University, Chiang Mai 50200, Thailand

**Keywords:** echocardiography, fetus, cardiac function, heart failure, myocardial performance, pressure overload, volume overload

## Abstract

Fetal heart failure (FHF) is a condition of inability of the fetal heart to deliver adequate blood flow for tissue perfusion in various organs, especially the brain, heart, liver and kidneys. FHF is associated with inadequate cardiac output, which is commonly encountered as the final outcome of several disorders and may lead to intrauterine fetal death or severe morbidity. Fetal echocardiography plays an important role in diagnosis of FHF as well as of the underlying causes. The main findings supporting the diagnosis of FHF include various signs of cardiac dysfunction, such as cardiomegaly, poor contractility, low cardiac output, increased central venous pressures, hydropic signs, and the findings of specific underlying disorders. This review will present a summary of the pathophysiology of fetal cardiac failure and practical points in fetal echocardiography for diagnosis of FHF, focusing on essential diagnostic techniques used in daily practice for evaluation of fetal cardiac function, such as myocardial performance index, arterial and systemic venous Doppler waveforms, shortening fraction, and cardiovascular profile score (CVPs), a combination of five echocardiographic markers indicative of fetal cardiovascular health. The common causes of FHF are reviewed and updated in detail, including fetal dysrhythmia, fetal anemia (e.g., alpha-thalassemia, parvovirus B19 infection, and twin anemia-polycythemia sequence), non-anemic volume load (e.g., twin-to-twin transfusion, arteriovenous malformations, and sacrococcygeal teratoma, etc.), increased afterload (intrauterine growth restriction and outflow tract obstruction, such as critical aortic stenosis), intrinsic myocardial disease (cardiomyopathies), congenital heart defects (Ebstein anomaly, hypoplastic heart, pulmonary stenosis with intact interventricular septum, etc.) and external cardiac compression. Understanding the pathophysiology and clinical courses of various etiologies of FHF can help physicians make prenatal diagnoses and serve as a guide for counseling, surveillance and management.

## 1. Introduction

Fetal heart failure (FHF) is defined as inadequate tissue oxygen perfusion caused by inadequate cardiac output secondary to any causes, leading to a series of complex cardiovascular adaptations to improve blood flow and oxygen perfusion in developing vital organs, especially the brain, heart, liver and kidneys. FHF is not a specific disease but, rather, a late consequence of several disorders and may lead to intrauterine fetal death or severe morbidity. The common causes of FHF are as follows: low cardiac output (cardiomyopathy, myocarditis, and arrhythmia), high cardiac output (arteriovenous malformations, twin-to-twin transfusion, and longstanding anemia), increased afterload (advanced growth restriction and obstructive cardiac defects) and external cardiac compression (pleural/pericardial effusions and intrathoracic tumors). Diagnosis of FHF is based on various fetal echocardiographic findings, depending on the severity of cardiac dysfunction, including cardiomegaly, poor contractility, low cardiac output, venous congestion, effusions, and typical findings of specific cardiac anomalies. Currently, fetal echocardiography can diagnose several forms of congenital heart anomaly and assess the prognosis based on their abnormalities and severity of cardiac dysfunction. Commonly used diagnostic techniques for assessment of FHF are shortening fraction, myocardial performance index, arterial and systemic venous Doppler waveforms, and cardiovascular profile score (CVPs). To enhance clinical practice, this review focuses on a straightforward method for rapid evaluation of a fetus with possible FHF. It is based on common echocardiographic examinations used in prenatal detection of fetal heart failure and common disorders associated with cardiac dysfunction that culminates in heart failure, together with their pathophysiology.

## 2. Methods of Review

A search strategy was used to identify peer-reviewed articles published between January 1990 and July 2022, using the following databases: PubMed, SCOPUS, and Web of Science. The article types included original research, reviews and professional guidelines. The authors independently assessed and validated the title, abstract, and full-text of the studies meeting the inclusion criteria: those specific to fetal heart failure. The key words were anemia, cardiac (heart) failure, cardiac function, fetus, high-output, low-output, prenatal diagnosis, and ultrasound.

The retrieved articles were firstly screened by abstracts to identify and select the articles focusing on the following topic criteria:***Techniques used for fetal hemodynamics*:** used for assessment of fetal heart function, focused on techniques which are simplified, practical (because of the clinical purpose), well evaluated in several studies with reproducibility and simplicity.***Prenatal diagnosis of FHF*:** To present consistent evidence of fetal hemodynamic changes in fetal heart failure secondary to different causes. (Note that this review emphasizes prenatal diagnosis based on fetal echocardiography, not treatment, which depends on the etiologies)***Common etiologies of FHF*:** Though a large number of disorders can cause FHF, this review assessed and validated only common causes relevant to clinical practice, including dysrhythmias, anemia, non-anemic volume load (twin-to-twin transfusion, arteriovenous malformation, etc.), increased afterload (fetal growth restriction and outflow tract obstruction), and cardiomyopathies.

***On data extraction and quality assessment:*** The retrieved full-text articles were subjectively assessed and validated. The articles were selected by agreement of the two raters (authors). The main insights extracted from the selected articles were evidence of pathophysiology, cardiovascular responses to different causes of FHF, and typical sonographic features in different causes.

**Key Message:** This review provides updated knowledge on various aspects of fetal heart failure, as follows:Update and summary of well-accepted useful techniques for assessment of fetal cardiac function in clinical practice.Pathophysiology of FHF secondary to different causes with different cardiovascular changes in the early phase, though prenatal features in the advanced stage of CHF are relatively similar.Illustration of various cardiovascular responses to different causes of FHF with demonstration of many scenarios to facilitate early detection, which is helpful in counseling, predicting outcomes, and effective surveillance or management.

## 3. Evolution of Fetal Heart Failure

In a normal fetus, the developing heart has high reserve potentials to cope with overloading and to maintain a stable combined cardiac output (CCO) of around 500 mL/kg/min. Notably, in the case of a cardiac defect of one ventricle, the other can maintain a normal CCO because of connecting parallel circulation [1,2,3]. In general, FHF is the consequence of a progressive increase in cardiac workload, secondary to any causes, either increased afterload or volume load, which needs a higher cardiac output to maintain tissue perfusion. However, irrespective of the underlying causes, longstanding volume load or a pressure load in one or both overworked ventricles cause myocardial wall stretching stress and eccentric remodeling and also decrease ejection fraction [3]. Additionally, ventricular overload causes a progressive increase in the size and number of cardiomyocytes [4,5], leading to concentric remodeling and hypertrophy, which result in an increase in myocardial oxygen consumption, vulnerability to myocardial fibrosis, lower compliance and worsening cardiac performance [3]. Pressure load and a reduction of circulating volume upregulate the renin–angiotensin system, causing local production of angiotensin II and tissue growth factor (TGF)-β1, facilitating fibrosis [6,7], leading to diastolic and systolic dysfunction. As a consequence, the end-diastolic filling pressure and central venous pressure increase to improve the cardiac output, thus facilitating atrial natriuretic peptide release, which promotes an increase in capillary permeability, thereby enhancing extravascular fluid shift, reduced lymphatic drainage, low CCO and development of hydrops [8].

When the compensatory mechanism becomes exhausted, myocardial performance is markedly reduced, resulting in critically low CCO, poor tissue perfusion, progressive acidosis and, eventually, fetal demise. The total body fluid volume and extracellular fluid proportion of fetuses are relatively much greater than those of adults. Clearance of extracellular fluid in a fetus mainly depends on highly effective lymphatic drainage. A fetus is very vulnerable to extracellular fluid shift, even in a milieu of minimal increase in systemic venous pressure. Accordingly, any condition with increased venous pressure and reduced lymphatic drainage can markedly enhance fluid shift, leading to hydrops fetalis, as commonly seen in advanced FHF. Based on studies in fetal sheep, venous pressure is directly related to the pressure gradient between the right atrium and the right ventricle during end-diastole [9,10]. Since fetal lymphatic flow is up to five times greater than in the adult, changes in systemic venous pressure have a much greater effect on lymphatic flow in the fetus than in the adult with cessation of flow at pressures of 15mmHg versus 25mmHg in the adult [11]. Venous pressure is also related to both ventricular compliance and ventricular end diastolic pressure. Thus, alterations in central venous blood velocity patterns accurately reflect abnormalities in cardiac hemodynamics [12,13,14]. Additionally, elevated filling pressures and decreased systemic blood pressure trigger hormonal responses such as release of vasopressin (decreased urinary production), angiotensin II (increased fluid accumulation), and atrial natriuretic peptide (increased capillary permeability), further favoring the development of hydrops [15,16,17]. 

In fact, hydrops fetalis can be a cause or consequence of heart failure. The mechanism for end-stage heart failure in hydrops fetalis may be summarized as follows: Reduced ventricular contraction leads to a decrease in compliance and the Frank Starling mechanism, resulting in increased dependence of cardiac output on heart rate. As a consequence, adenoreceptors are decreased and facilitate cardiac failure [18]. However, it should be kept in mind that hydrops fetalis is not always caused by FHF. For example, in fetal anemia, hypervolemia, a compensatory mechanism to increase cardiac output to maintain tissue perfusion, can cause minimal increase in venous pressure and lead to hydrops in spite of the absence of significant heart failure [19,20].

## 4. Prenatal Assessment of Fetal Hemodynamics

Clinically, FHF, regardless of the underlying cause, is usually encountered unexpectedly in most cases at the time of routine fetal ultrasound examination of asymptomatic mothers. Nevertheless, in several cases, fetal echocardiography is performed because of a higher risk of FHF, such as advanced fetal growth restriction, decreased fetal movements and polyhydramnios. Therefore, prenatal diagnosis of FHF is usually based on echocardiographic assessment of cardiac dysfunction, after the detection of abnormal findings on routine obstetric ultrasound, such as abnormal heart rates, structural heart defects, and hydropic signs. However, it should be realized that hydrops can be a consequence of FHF but is a nonspecific finding, since hydropic changes are more commonly associated with extra-cardiac causes and chromosomal disorders, in the absence of a major cardiac condition [21,22], or anemia-related hydrops with good cardiac compensation [19,23,24]. Echocardiographic assessment of fetal cardiac function is the most important part in the detection of dysfunction and diagnosis of FHF. A variety of echocardiographic techniques can be used to detect FHF and predict the outcomes. In practice, the assessment is usually accomplished using conventional two-dimensional ultrasound, Doppler, and M-mode echocardiography, possibly successfully performed even as early as late first trimester [25]. The commonly used methods for the evaluation of fetal cardiac function are described below [26,27,28,29] and summarized in Table 1. Based on this review, for clinical purposes in actual practice, fetal cardiovascular assessment for FHF should include cardiac size, hydropic signs, umbilical artery (UA) Doppler, myocardial performance index (MPI) or modified Tei index, E/A waveform analysis, the presence of atrioventricular valve (AV) regurgitation, shortening fraction, middle cerebral artery (MCA) Doppler, ductus venosus (DV) waveforms, and umbilical vein (UV).

### 4.1. Ventricular Shortening Fraction (SF)

SF is one of the most commonly used M-mode parameters to quantify the global fetal systolic ventricular function using measurements of the percentage changes in ventricular dimensions at the ends of diastole (EDD) and systole (ESD), respectively [30]. The measurement is made on the standard four-chamber view, at the greatest dimension, typically at or just below the level of the atrioventricular (AV) valves. SF is calculated as follows: Left ventricle shortening fraction (LVSF) and right ventricle shortening fraction (RVSF) are calculated using this equation: SF = Dimension at end-diastole—Dimension at end-systole/Dimension at end-diastole and presented as percentage. Normal RVSF and LVSF are greater than 28% [31]. A lower SF indicates decreased systolic function, commonly seen in myocardial disease, an abnormally high afterload.

### 4.2. Fetal Cardiac Size 

Cardiothoracic area ratio (CTR), measured on the standard four-chamber view, is the most commonly used parameter for evaluation of cardiac size. A normal cardiac area is less than 35% of the thoracic area, measured on the same plane. Measurements of the ventricular areas of both ventricles and ventricular wall thickness are also helpful. The values confined within +2 SDs of the reference ranges are considered normal [32].

Cardiothoracic diameter ratio is also extensively used in predicting fetal anemia, especially fetal Hb Bart’s disease [24,33,34,35]. The measurement is performed on the standard four-chamber view, similar to area measurement, at the greatest dimension during the end-diastole. Typically, a value greater than 0.5 is used as a cut-off at mid-pregnancy. However, the ratio is gestational age-dependent; thus, it should be interpreted by comparison with the nomogram. The reference ranges of cardiothoracic diameter ratio are available for clinical use [36,37].

### 4.3. Fetal Valve Competency 

Color flow mapping and spectral Doppler study of the ventricular inflow and outflow are used to detect and quantify the degree of valve regurgitation, respectively. Holosystolic regurgitation of the tricuspid valve (TV), occurring during the entire duration of systole, is commonly associated with increased RV pressure, possibly related to outflow obstruction or arterial hypertension and also associated with structural cardiac defects, such as the Ebstein anomaly, TV dysplasia, etc. [38,39,40,41]. Trivial regurgitation of TV, non-holosystolic with a maximum velocity of less than 200 cm/s, is considered as normal variation [42,43]. Regurgitation of mitral valves (MV) should always be considered abnormal.

### 4.4. Cardiac Output 

Measurement of cardiac output (CO) can be used for quantitative assessment of LV or RV ventricular function or both by combined cardiac output (CCO). CO can directly represent ventricular systolic function. A decrease in CO, defined as a Z-score of less than 2, is indicative of reduced ventricular filling or contractility, found in low output failure, which accounts for the majority of cases with FHF, including structural cardiac defects, [39,40,41,44] cardiomyopathies [45], and arrhythmias [46,47,48]. An increase in CO, a Z-score of greater than two, is indicative of reduced afterload, such as in fetal anemia [49]; volume loading, such as in AV shunting [50,51]; highly vascularized masses (sacrococcygeal teratoma, acardiac twin, chorioangioma and lung sequestration) [52,53,54,55,56] and agenesis of the ductus venosus [57].

CO can be calculated by the following equation:*CO (milliliters per minute)* = *VTI* × *π* × (*D*^2^/4) × *FHR*

*(VTI:* velocity-time integral of the systolic aortic or pulmonary flow; *D:* annulus diameter of the corresponding valve; and *FHR:* fetal heart rate)

To obtain an accurate VTI, the Doppler sampling cursor must be placed just immediately distal of the aortic or pulmonary valves, with an ultrasound beam insonation angle of less than 15–20° or with the beam as parallel as possible to the direction of blood flow. The diameters of the semilunar valves are measured on the long axis zoom-enlarged view, with cursor placement from inner wall to inner wall at the level of the valve insertion immediately after the valve opening, with the best frame selection by cine loop replay. Many reference ranges of fetal cardiac outputs are available for clinical use [58].

### 4.5. The Myocardial Performance Index (MPI) 

The MPI or Tei index (Figure 1), introduced by Tei et al. [59] in 1995, is a combined measure of global ventricular function, calculated as the ratio of the sum of the tricuspid or mitral isovolumetric contraction time (ICT) and isovolumetric relaxation time (IRT) to the duration of the pulmonary or aortic ejection time (ET), for the RV and LV, respectively. Normal MPIs of the LV and RV are <0.48 [60]. In 2005, Andrade et al. [61] proposed the modified MPI, involving an assessment of the movements (clicks) of the mitral valve (MV) and aortic valve (AV). They demonstrated the improvement of intra- and interobserver agreement compared to the original MPI method. Currently, the modified MPI is considered as a more reliable and useful tool in evaluation of fetal cardiac function in various fetal conditions at risk of evolution of cardiac dysfunction or FHF, for example fetal anemia, fetal growth restriction, twin-twin transfusion syndrome, and maternal diabetes mellitus. Additionally, prolonged IRT, as a component of MPI, suggests diastolic dysfunction (normal LV IRT < 43 milliseconds) [62]. The measurement of MPI in the fetus is relatively simple, though specific training is required; however, it cannot effectively distinguish between systolic and diastolic function and is affected by ventricular loading [63]. Additionally, some limitations in clinical translation, such as a poorly standardized technique with variations, ultrasound machine settings, cursor placement, and need of training, can result in significantly different MPI values [64]. Many reference ranges of fetal MPI are available for clinical use [27,65,66].

### 4.6. Ventricular Inflow 

Fetal diastolic function can be evaluated by cardiac Doppler waveforms, including isovolumetric relaxation and characteristics of E–A waves of the inflow. Doppler waveforms across the atrioventricular valves are biphasic, with a dominant A-wave. With advancing gestation, ventricular compliance is increased, resulting in a progressive increase of the E-wave, which leads to an E/A ratio of close to 1. A shorter or monophasic inflow pattern often suggests abnormal ventricular filling or diastolic dysfunction (normal ventricular inflow time > 38% of the cardiac cycle length [67]. The inflow time or filling time fraction (FTF) is useful in evaluation of fetal cardiac diastolic dysfunction [68,69]. FTF was calculated by dividing the inflow time by the cycle time and multiplying by 100. Inflow time is the interval between the beginning of the E-wave (the rising point of the waveform from the baseline, typically seen as a small echogenic click of the AV valve opening at the onset) and the end of the A-wave (the termination point on the baseline at the beginning of the AV valve closure click). The reference ranges of FTF are currently available [68].

### 4.7. Systemic Venous Pressure 

The Doppler venous waveforms are triphasic in shape, with the first phase corresponding to atrial diastole and ventricular systole, the second phase corresponding to early diastole, and the third phase corresponding to late diastole or the atrial contraction [9,12]. Diastole begins with the opening of the AV valves, causing inflow across the AV valves that are consistent with an E-wave with an acute decrease in atrial pressure and the simultaneous occurrence of a D-wave of venous flow. Then, early diastolic venous flow slowly decreases with increasing ventricular pressure during ventricular filling. Once the atria contract in late diastole, producing an A-wave of inflow across the AV valves, there is an acute increase in atrial pressure, which slows down the forward flow in the ductus venosus and pulmonary veins and reverses the venous flow in the hepatic veins and the inferior vena cava (IVC) (reversed A-wave). Venous flow during normal RV diastolic function is characterized by: (1) positive A-wave in the ductus venosus; (2) A-wave reversal of <20 cm/s in the IVC [26]; and (3) no pulsation in the umbilical vein during the entire cardiac cycle. Increased systemic venous pressure, reflecting diastolic dysfunction, can also be evaluated by analyzing the Doppler waveforms of the IVC and the ductus venosus as well as the pulmonary vein for left atrial pressure, which typically are triphasic in appearance. In normal fetuses, the flow during late diastole, reflected by A-waves, is reversed in the IVC [70], whereas the forward flow is present during the entire cardiac cycle in the ductus venosus [71]. The flow velocity waveforms of the pulmonary veins, similarly to those in the ductus venosus, reflect pressure changes in the left atrium during the cardiac cycle.

### 4.8. The Cardiovascular Profile Score (CVPS)

CVPS is a composite scoring system, first reported by Huhta et al. [8], that has been used to characterize and grade as well as follow up on the severity of FHF. It is based on assigning two points for each of five echocardiographic categories, as presented in Table 2: (1) fetal effusions, (2) venous Doppler findings, (3) cardiac size, (4) cardiac function, and (5) arterial Doppler findings. Severity of FHF is rated on a 10-point scale, with a score of 10 considered as the absence of heart failure. Points are deducted for abnormalities of each component marker. The severity of heart failure is graded according to CVPS as follows [72,73]: mild (8 or 9 points), moderate (6 or 7 points), and severe (<5 points). One point is deducted from the CVPS for each of the following findings: (1) ascites or pleural effusion or pericardial effusion, (2) reversed flow in the ductus venosus, (3) cardiomegaly with a cardio-thoracic ratio (CTR) between 35% and 50%, (4) holosystolic TR or ventricular shortening fraction < 28%, and (5) AEDV in the umbilical artery. Two points are deducted for each of the following markers: (1) fetal skin edema, (2) pulsatile umbilical vein flow, (3) cardiomegaly with a CTR > 50%, (4) holosystolic mitral regurgitation or dP/dt < 400 mm Hg/s or monophasic inflow pattern, and (5) REDV in the umbilical artery. Of the five categories, reversed end-diastolic flow of the ductus venosus and umbilical vein pulsations are the best predictors of adverse outcome [72]. CVPS has been validated as a predictor of severe fetal and neonatal morbidity and mortality in several studies [73,74,75,76]. The severity of heart failure may be graded according to CVPS as follows: mild (8 points), moderate (6 or 7 points), and severe (5 points or less).

***Other modalities*:** Several echocardiographic parameters of fetal cardiovascular disorders have been shown to correlate with adverse perinatal outcomes [29]. Several modalities seem to be helpful in the detection of the signs of fetal cardiac dysfunction in preliminary studies [77,78,79,80]. However, such new techniques need to be elucidated in the aspects of the true relationship with the prognosis and their reproducibility. For example, Huang et al. [80] showed that with speckle-tracking, fetuses exposed to GDM may show cardiac dysfunction before the onset of cardiac morphologic abnormalities, and the right ventricle is more vulnerable than the left during fetal development. Guo et al. [81] studied annular plane systolic excursion and showed that free-angle M-mode (FAM) of mitral annular plane systolic excursion (MAPSE)and FAM of annular plane systolic excursion (APSE) Z-scores could be markers for assessing heart systolic function and severity in FHF. Some studies showed that tissue Doppler is more accurate than spectral Doppler and enables the segmental analysis of diastolic myocardial function [29,65,82].

## 5. Clinical and Echographic Features of Different Forms of FHF

Fetal dysrhythmiaFetal anemia (e.g., alpha-thalassemia, parviovirus B19, twin anemia- polycythemia sequence: TAPS)Non-anemic volume load (twin-twin transfusion (recipient twin) arteriovenous malformations, sacrococcygeal teratoma agenesis of ductus venosus, aneurysm of vein of Galen, etc.)Increased afterload (intrauterine growth restriction, outflow tract obstruction such as critical aortic stenosis)Intrinsic myocardial disease (cardiomyopathies)Congenital heart defects (Ebstein anomaly, hypoplastic heart, pulmonary stenosis with intact interventricular septum, etc.)External cardiac compression

### 5.1. Fetal Dysrhythmias 

It has long been known that persistent fetal dysrhythmia can be associated with FHF. However, in recent times, there has been a decline in such association because of earlier detection and intrauterine treatment before the development of FHF [83]. 

Supraventricular tachycardia and atrial flutter; the most common causes in this group, usually associated with tachycardia of greater than 220 bpmAV block with cardiac defects; commonly associated with left atrial isomerism and corrected transposition of the great arteriesAV block without congenital heart defects, commonly associated with SSA/SSB (Ro/La) antibodies or idiopathicAccelerated junctional or ventricular rhythm, relatively rare

*Supraventricular tachycardia (SVT)* (Figure 2) and atrial flutter are the most common causes of fetal tachycardia, accounting for 66 to 90% of all cases and also constituting the most common fetal arrhythmia associated with the evolution of FHF [47,84]. In SVT, the tachycardia range is approximately 220 to 240 bpm with 1:1 ratio of AV conduction and loss of fetal heart rate variability. Atrial flutter is defined as a rapid regular atrial rate of 300 to 600 bpm, accompanied by a variable degree of AV conduction block, resulting in a slower ventricular rate, approximately 220 to 240 bpm. Ventricular tachycardia presents with ventricular rates of more than 180 bpm in the setting of AV dissociation. Though it can be associated with FHF [85], it is a rare cause.

The mechanisms responsible for heart failure caused by SVT may be associated with increased ventricular and atrial pressures and myocardial energy loss, leading to cardiac dysfunction and culminating in FHF. Very rapid heartbeat directly decreases ventricular filling and ejection time, causing an increase in ventricular and atrial pressures. Gembruch et al. [86] demonstrated that in human fetuses with SVT (greater than 210 bpm), systemic venous pressure was significantly increased, as indicated by an increase in the reversal of A-waves of venous Doppler flow patterns in the IVC and the ductus venosus, as a consequence of increasing reversal of flow in atrial systole.

Using an animal model, Schmidt et al. [87] demonstrated that rapid atrial pacing, representing SVT, resulted in rapid worsening of systolic and diastolic function. Such deterioration developed simultaneously with decreasing fetal myocardial glycogen storage, which is an energy substrate for the immature heart, probably contributing to the myocardial dysfunction [87]. The authors also showed that maternal infusion of insulin and glucose could result in significant improvement in systolic and diastolic function, indicative of the contribution of a reduced myocardial glycogen store in the evolution of CHF [88]. The risk factors of SVT that are linked to the evolution of FHF include occurrence at an earlier gestational age, an association with a structural heart defect, and a more incessant nature [89].

*Fetal bradycardia*, defined as a sustained fetal heart rate of less than 100 bpm, is commonly caused by sinus bradycardia, blocked atrial bigeminy/trigeminy, or a high-degree AV block. Of the bradycardia group, a complete atrioventricular block (CAVB) (Figure 3) is the most common entity in association with the evolution of FHF. Approximately 50% of CAVB cases occur in association with maternal autoantibodies related to connective tissue diseases; 40–50% of CAVB cases occur in fetuses with congenital heart defects, and the remaining are of unclear etiology (4–10% of cases) [90].

Fetuses with an AV block make attempts to increase compensatory ventricular stroke volumes to maintain adequate cardiac output. Nevertheless, some cannot tolerate such excessive workload, culminating in FHF and, subsequently, fetal demise. FHF and fetal demise associated with AV block are more common in cases with a ventricular rate of less than 55 bpm, which increases the risk of fetal demise and should be considered as a poor prognostic factor [90,91]. Additionally, several studies have shown that fetal heart block in the presence of a structural cardiac defect is associated with a very high perinatal mortality rate [90,92]. Additionally, coexistent myocardial disease with cardiac dysfunction, found in 15–25% of cases with maternal autoantibody-induced AV block [93] and observed in left atrial isomerism, is associated with the development of FHF and fetal demise [92]. In mothers at high risk of having a child with CAVB, the use of HCQ may protect against recurrence of disease in a subsequent pregnancy.

### 5.2. Fetal Anemia 

Fetal anemia either caused by alpha-thalassemia, isoimmunization or parvovirus B19 infection can be associated with hydrops fetalis. Fetal anemia may be a manifestation of twin-anemia-polycythemia sequence, seen in 3–5% of monochorionic twin pregnancies [94]. In Southeast Asia, fetal Hb Bart’s disease is the most common cause of hydrops fetalis, whereas isoimmunization and parvovirus B19 are commonly reported from the Western world. However, fetal anemia, in fact, rarely cause FHF, as believed in the past. Actually, a volume load secondary to increased cardiac output in fetuses with anemia can cause shifting of fluid from the vascular compartment to the interstitial space of the fetal body, in spite of the absence of CHF [23]. FHF secondary to fetal anemia developed only in the late stage of longstanding severe anemic hypoxia, usually long after the development of hydrops fetalis. Based on our extensive experience in fetal Hb Bart’s disease, fetal response to anemia can be summarized as follows [23]: (1) Morphologically, the earliest sonographic sign during the pre-hydropic phase is an increase in cardiac size [24,33,35] as well as a more globular shape or increased sphericity index, followed by several other signs, such as placental enlargement, hepatomegaly, splenomegaly, etc. (2) Concerning hemodynamic changes, the earliest sonographic sign is an increase in middle cerebral artery–peak systolic velocity (MCA–PSV) and those of other fetal arteries, such as the splenic artery. (3) In fetal anemia, the heart works harder or is overworked to increase cardiac output, when compared to a normal fetal heart. Nevertheless, the fetal heart has very high reserve potentials to combat anemic hypoxia. In the early phase, the heart has a good performance in spite of the workload, without increased central venous pressure or preload index. The fetuses show an increase in the Tei index (MPI) without compromising contractility. The hard-working phase with good performance may last long, until late gestation when the adaptive mechanism of the heart becomes exhausted and then heart failure develops. (4) The overworked heart gradually develops myocardial cellular damage, in spite of good cardiac performance. (5) Hydrops fetalis secondary to anemia is primarily caused by volume load together with high vascular permeability during fetal life and hypoalbuminemia. It should be emphasized that hydrops fetalis in fetuses with anemia is not a sign of FHF in most cases. The prognosis of this condition is much better than that caused by FHF. (6) FHF is a very late consequence of anemia, developing only when the adaptive mechanism becomes exhausted, long after the occurrence of hydrops fetalis. In other words, FHF is a very late consequence of a longstanding overworked heart, and it is likely to potentiate hydropic changes to become more severe in the late state. On the contrary, hydrops fetalis itself may accelerate the heart failure by external cardiac compression. (7) Fetal echocardiography is highly effective in identifying fetal anemia in the pre-hydropic phase and in differentiating FHF secondary to anemia from that of non-anemic causes. Increase in fetal cardiac size and MCA-PSV are very useful in the detection of fetuses with anemia in the pre-hydropic phase. (8) Though, theoretically, intrauterine blood transfusion (IUT) for fetal anemia should be performed before the development of anemic hypoxia in the pre-hydropic phase, IUT even in cases of hydropic changes can also result in good outcomes as long as FHF has not yet developed [95].

Of note, fetuses have high reserve potentials in responding to anemia. In spite of increased volume load, central venous pressure is not increased during the compensatory state, as indicated by increased forward flow in the ductus venosus during atrial contraction (markedly increased positive A-wave) (Figure 4) [19]. Several cases of fetuses with anemia show prolonged ICT [96], more pronounced than prolonged IRT as commonly seen in most cases developing FHF from other causes.

We previously pointed out that based on our extensive experience in fetal anemia secondary to Hb Bart’s disease, fetal hemodynamic response to anemia is consistent with the evidence of an association between heart failure and anemia in adult life. In postnatal or adult life, anemia is commonly quoted as a leading cause of high-output heart failure, as seen in patients with beta-thalassemia diseases [97,98]. However, the pathophysiology of heart failure caused by anemia is still unclear. Actually, anemia, even in severe cases, rarely causes heart failure. Moreover, patients with heart failure associated with anemia are likely to have high-output failure superimposed upon other underlying cardiac disorders, such as valvular heart defects or preexisting left ventricular dysfunction [99]. It has been demonstrated that in patients with Hb levels of as low as 7 g/dL, normal cardiac hemodynamics could be maintained [100]. When Hb levels are decreased to 5–7 g/dL, the cardiac output is increased, but heart failure does not develop. Importantly, heart failure develops in the absence of underlying cardiac disorders only when Hb levels of less than 5 g/dL are reached. Additionally, in adults, chronic anemia can also accelerate the development of heart failure secondary to other causes. It has been demonstrated that chronic anemia is likely to worsen the severity of heart failure, but anemia itself is often associated with several comorbidities. It is unclear whether anemia is an independent factor of failure progression, overall disease severity or comorbidities that contribute to failure progression.

In fetal anemia, the oxygen-carrying capacity is reduced, leading to a decrease in systemic vascular resistance and an increase in cardiac output and widening of the arteriovenous O2 difference, as a compensatory mechanism [101,102]. Therefore, severe anemia can cause volume overload and increased stroke volume, which can alter ventricular function [103]. According to studies involving adaptive changes in post-transfusion anemic fetuses and studies on fetal lamb models, in fetuses with anemia, longitudinal strain, cardiac output, and shortening fraction are increased prior to intrauterine transfusion (IUT), when compared to the controls. Immediately after IUT, all parameters of systolic function are reduced to below the control levels, and 24 h after IUT, cardiac output, shortening fraction, and longitudinal strain are comparable with those of the controls [104,105,106]. This evidence suggests that a decrease in afterload is a primary mechanism of compensation in anemia, whereas an IUT increases intravascular volume and, therefore, flow resistance. A decrease in afterload caused by a reduction of serum viscosity may also play a role [103]. Additionally, anemia may cause peripheral vasodilation [107] associated with endothelial dysfunction [108]. In some fetuses with longstanding anemia, ventricular hypertrophy has also been detected in association with elevated catecholamines released in response to anemic hypoxia [109]. Nevertheless, ventricular hypertrophy is relatively rare in fetuses with anemia. Of note, cardiac compromise is more often seen in anemia caused by parvovirus-B19 infection, likely related to coexistent myocarditis [110]. Additionally, reduced colloid oncotic pressure, hypoxia-induced capillary damage and, possibly, portal hypertension can contribute to hydrops development [19]. As mentioned earlier, anemia-associated hydrops fetalis is not a sign of FHF in most cases.

Treatment of fetal anemia depends on the underlying causes. For example, termination of pregnancy is usually offered for lethal conditions such as fetal Hb Bart’s disease, whereas IUT is recommended in most cases of severe anemia secondary to parvovirus B19, Rh isoimmunization or Hb H hydrops fetalis. Current guidelines recommend cordocentesis to measure fetal Hb in case of MCA-PSV greater than 1.5 MoM and IUT if fetal hematocrit is less than 30% [111].

### 5.3. Non-Anemic High Cardiac Output

Several disorders are associated with an increase in combined cardiac output, such as arteriovenous malformations, the recipient of twin-to-twin transfusion syndrome and twin reversed arterial perfusion (TRAP) sequence (Figure 5), in which an acardiac twin without a heart or with a rudimentary heart is perfused by a pump twin [26]. Arteriovenous malformations most commonly occur in a highly vascularized mass, such as a placental chorioangioma, sacrococcygeal teratoma (Figure 6), hepatic hemangioma and aneurysm of vein of Galen. In these conditions, left to right shunting in the lesions causes systemic volume overload and is associated with increasing ventricular and atrial filling pressures. 

The severity of cardiac overload is directly related to the volume of shunting, usually directly correlated with the tumor size or the acardiac twin size relative to fetal weight [112,113]. The cases with higher shunting volume or huge tumor mass have a high perinatal morbidity and mortality rate. As consequences of volume overload, the fetuses develop global cardiomegaly, increased cardiac output and cardiac dysfunction in varying degrees of severity. Typically, to cope with the cardiac overworking, compensatory afterload reduction occurs. However, when compensatory afterload reduction is overwhelmed, AV valve insufficiency or low CVPs can be expected. The measurement of combined cardiac output may be used to predict fetal compromise [114] and to guide the need for intrauterine intervention. It has been suggested that combined cardiac outputs of two-fold or greater of the reference range may be a cut-off for the development of FHF in sacrococcygeal teratoma or agenesis of the ductus venosus [115,116]. Fetal hydropic changes or fluid collection in the body cavities may develop in severe cases [117], though cardiac contractility or shortening fraction may be normal, hyperdynamic, or poor in advanced stages. Fetal echocardiography usually shows some degree of cardiac compromise, especially an increase in cardiac output [118,119] and a lower cardiovascular profile score (CVPs) [120], as well as abnormalities of venous Doppler waveforms. These fetal echocardiographic findings are also very helpful as guidance for prenatal intervention, such as obliteration or embolization of the main vessels feeding the tumor or perivascular sclerosis, surgical debulking of the tumor, arterial ligation, radiofrequency ablation of vessels of an acardiac twin, etc. Additionally, serial assessment of cardiac function in less severe cases is necessary for early detection of cardiac dysfunction or decompensation. The useful parameters for assessment include cardiac size, ventricular dilatation, combined cardiac output, spectral Doppler flow of the IVC, the ductus venosus or the umbilical vein as well as severity of AV valve regurgitation. Of note, different from volume load caused by anemia in which low viscosity and a marked decrease in afterload play an important role in the reduction of cardiac pressure load, the volume load in fetuses with non-anemic high output is associated with more rapid and severe cardiac pressure load and is more likely to cause FHF than anemic causes.

#### Increased Afterload

Increased ventricular afterload can eventually result in FHF, when compensatory mechanisms become exhausted. The causes of increased afterload are as follows: (1) blood flow obstruction in the heart, such as aortic stenosis, pulmonary stenosis, restrictive foramen ovale and constriction of the ductus arteriosus. (2) Extra-cardiac causes, including increased systemic vascular resistance caused by placental insufficiency as commonly seen in fetal growth restriction and twin-twin transfusion syndrome (TTTS).

### 5.4. Twin-to-Twin Transfusion Syndrome (TTTS)

TTTS is one of the most recognized causes of elevated afterload in the recipient twin. TTTS occurs in approximately 15–20% of monochorionic twin pregnancies. Pathologically, TTTS is caused by imbalance of blood flow through placental vascular anastomoses between the twins, with umbilical/chorionic arteries of the donor perfusing deep into the cotyledons of the recipient. In TTTS, the twins develop discrepancies in growth, blood volume, amniotic fluid volume and cardiac size. While the donor is typically small-for-date with oligohydramnios, the recipient suffers from volume overload. Because of hemodynamic imbalance, the donor has hypovolemia, while the recipient has volume overload, leading to a discrepancy in the fetal hemodynamic responses of both twins. The recipient twin shows an increase in afterload in response to volume load by releasing vasoactive peptides, probably originating from the placenta. The blood levels of several vasoconstrictors, such as endothelin-1, renin, and angiotensin- II, are markedly increased in the recipient twin [121], contributing to an increase in afterload. However, the donor twin may have high systemic vascular and placental resistance because of a compensatory response to a decrease in intravascular volume, but less alteration in cardiac function is often observed.

On fetal echocardiography, the recipient twin typically shows enlarged cardiac size; progressive biventricular hypertrophy, more pronounced on the right ventricle; elevated systemic pressures, leading to significant tricuspid and mitral regurgitation [122]; diastolic dysfunction with shortened filling fraction time; monophasic inflow of the E-A waveforms; increased Tei index primarily due to prolonged IRT [123]; and abnormal Doppler flow in the ductus venosus and IVC. Both diastolic and systolic dysfunction tend to be progressive and eventually result in FHF. The recipient twin has decreased ventricular strains of both sides [124,125], whereas the donor twin often shows a reduced strain rate of the right ventricle [125] but an increased left ventricular strain rate, likely caused by lower cerebrovascular resistance; however, placental resistance is increased. Differential circulation in the donor contributes to an increase in brain diastolic blood flow or cephalization, previously known as brain sparing, also associated with the response of myocardial deformation to alterations in differential afterload [125].

### 5.5. Fetal Growth Restriction (FGR)

Normally structural fetuses with growth restriction are commonly caused by uteroplacental insufficiency (UPI) which is closely associated with increased placental resistance, resulting in reduced end-diastolic Doppler velocity in the umbilical artery and increased cardiac afterload. 

UPI results in chronic fetal hypoxia, which affects cardiac function with various mechanisms in FGR. In the early onset, FGR adaptive mechanisms involve the diversion of the cardiac output preferentially in favor of the brain (cephalization) and the heart. In the case of further worsening of fetal hypoxia, in addition to abnormal arterial flow, cardiac dysfunction and abnormal venous flow eventually occurs in order. In the uterine artery ligation rat model of FGR, growth-restricted fetal hearts had reduced wall thickness-to-diameter ratio, indicating LV dilatation, and they had elevated E/A ratios and a reduced ventricular shortening fraction, suggesting systolic and diastolic dysfunction, similar to human FGR [126].

In UPI-associated FGR (Figure 7), placental resistance is increased, indicated by an increase in systolic/diastolic ratio or pulsatility index of the umbilical artery or AEDV/REDV in severe cases. Increased placental resistance results in cardiac afterload, leading to redistribution of fetal hemodynamics. Because blood flow to the placenta, mostly running through the right ventricle, pulmonary artery and ductus arteriosus, is more difficult due to high resistance, blood circulation in the right atrium preferably crosses the foramen ovale, passing through the left atrium, the left ventricle, and the aorta to perfuse the brain, leading to an increase in cerebral blood flow, known as cephalization or brain-sparing effect. End-diastolic flow in the middle cerebral artery (MCA) is increased, resulting in a decreased MCA-PI/UA-PI ratio. In FGR, because of longstanding exposure to high cardiac afterload, right ventricular hypertrophy and cardiac dysfunction subsequently occur. Typically, when the compensatory mechanism coping with high afterload becomes overwhelmed, an abnormally high Tei index, poor shortening fraction and low cardiac output occur, followed by an increase in systemic venous pressure or increased preload index in the ductus venosus and IVC, represented by abnormally pronounced A-waves. Finally, when cardiac decompensation is more obvious, cardiac Doppler pulsation signals are strong enough to transmit through the ductus venosus to the umbilical vein, leading to the development of umbilical vein pulsations, which is commonly seen in the late phase of cardiac decompensation. 

In response to chronic increased afterload, ventricular hypertrophy is commonly present in FGR [127]. Additionally, nearly half of the fetuses show sonographic signs of cardiac dysfunction. Additionally, cardiomegaly, monophasic inflow of the EA waveforms and holosystolic tricuspid regurgitation have been shown to be associated with adverse outcomes [74]. A study about cardiac function in FGR, assessed by cardio-STIC, demonstrated that reduced stroke volume occurred at the initial stage of fetal deterioration before the development of abnormal EF in FGR fetuses [128]. The authors suggest that stroke volume could be a sensitive indicator of cardiac dysfunction and is possibly better than EF measurement. In twin pregnancy, selective intrauterine growth restriction may occur in one twin in the absence of any other sonographic features of TTTS. As seen in FGR of singleton pregnancy, ventricular hypertrophy with dysfunction may develop and progress to FHF or even hydrops fetalis [129].

### 5.6. Critical Aortic Stenosis

Aortic stenosis (Figure 8) causes increased left ventricular afterload (leading to left ventricular systolic and diastolic dysfunction), ventricular growth failure (ending up with poor function or non-function) or left ventricular failure and development of endocardial fibroelastosis.

On echocardiography, typical features of critical aortic stenosis are as follows: an abnormal four-chamber view with left ventricular dilation, poor contractility or systolic dysfunction of the dilated left ventricle. The apex may still be formed by the dilated left ventricle, and the left atrium may be dilated because of mitral valve regurgitation. Pulsed Doppler shows forward flow across patent but stenotic aortic valve with high peak systolic velocity (generally greater than 200 cm/s), reversed flow in the aortic arch, reversed flow crossing the foramen ovale and monophasic and shortened mitral valve inflow time, indicating poor compliance of the ventricular wall and increased end-systolic pressure [130]. Reduction in aortic peak systolic velocity on follow-up ultrasound is a sign of progressive dysfunction of the left ventricle.

Bright echogenicity of the endocardium, reflecting endocardial fibroelastosis, is commonly seen in critical aortic stenosis and is associated with failure of biventricular repair [131].

Intrauterine intervention to relieve the aortic valve obstruction can improve both systolic and diastolic function and increase ejection fraction and mitral valve inflow time [132]. Additionally, RV-to-LV length ratio combined with LV pressure estimates, prior to fetal aortic valvuloplasty, may be used to predict a successful biventricular outcome with high sensitivity and specificity [133].

### 5.7. Ductal Arteriosus Constriction

Ductus arteriosus constriction (DAC) is most commonly associated with the use of nonsteroidal anti-inflammatory medications, especially indomethacin, in the late middle to early third trimester [38]. The constriction can cause an acute increase in right ventricular afterload, culminating in heart failure. Nevertheless, in cases of indomethacin-induced constriction, stopping such a therapy can reverse the constriction of the ductus within days.

On fetal echocardiography, the four-chamber view usually shows right ventricular hypertrophy and dilatation with impaired systolic function in many cases. Tricuspid regurgitation and right atrium enlargement is commonly visualized, reflecting elevated right ventricular systolic pressure. In acute cases, poor contractility of the right ventricle or ejection fraction may be demonstrated, likely caused by acute afterload elevation without compensatory myocardial hypertrophy. Pulsed Doppler shows high peak systolic velocities in the constricted ductus arteriosus, approximately 200–300 cm/s, as well as high diastolic velocities with a ductal pulsatility index of less than 1.9 [134]. The Tei index of the right ventricle is increased, associated with shortened ejection time, and returns to baseline coincident with ductal relaxation [135]. Monophasic inflow waveforms of the right ventricle are commonly seen. Tricuspid regurgitation and right atrium enlargement can predict the immediate prognosis of the newborn and provide guidance for the clinical judgment of the timing of pregnancy termination [136].

#### 5.7.1. Intrinsic Contractile Dysfunction

Cardiomyopathy refers to a wide range of genetic, metabolic, familial, and inflammatory myocardial disorders in the absence of a structural cardiac defect in most cases. It is a disease of the myocardium affecting the LV, RV, or both and is commonly associated with abnormal cardiac function. It is relatively rare, accounting for 2.5% of fetal heart disease [137]. Based on echocardiography, cardiomyopathy is typically classified into two phenotypes; hypertrophic and dilated cardiomyopathy. The etiologies of both phenotypes are idiopathic in most cases (50%), whereas some are associated with genetic and inflammatory diseases. The most well-known disorder found in association with a hypertrophic cardiomyopathy is maternal diabetes mellitus. Dilated/non-hypertrophic cardiomyopathy, representing two-thirds of cases, is generally recognized as an enlarged heart with a dilated left ventricle, right ventricle, or commonly both ventricles. Echocardiography shows a decrease in left or right ventricular shortening fraction in most cases and a normal septal thickness. Ventricular dilation is demonstrated in less than 25% of cases. Color or pulsed Doppler shows some degree of regurgitation of the valve of the affected ventricle.

Hypertrophic cardiomyopathy (Figure 9), accounting for one-third of cases, is generally recognized as an enlarged heart in association with ventricular wall hypertrophy of one or, generally, both ventricles. Echocardiographic features are cardiomegaly or increased cardiothoracic ratio (80% of cases) with increased ventricular myocardial thickness, defined as a Z-score of greater than 2, and a normal left and right ventricular end-diastolic diameter in the absence of anatomical or systemic diseases.

The prognosis of cardiomyopathy depends on the underlying associated abnormalities. Generally, a poor prognosis may be expected in cases of poor diastolic dysfunction such as monophasic inflow, umbilical vein pulsations, and hydrops fetalis [93,137]. Non-hypertrophic cardiomyopathy appears to have a better prognosis than hypertrophic cardiomyopathy. However, ventricular hypertrophy associated with maternal diabetes has a better prognosis [93].

#### 5.7.2. Congenital Heart Defect

Though congenital heart defects (CHD) are common, most of them are not associated with FHF. Approximately 5% of CHD are associated with heart failure [138] or CVPs of 7 or less [72]. Common CHD associated with cardiac failure are as follows [139]:Critical aortic stenosisEbstein anomalySevere atrioventricular valve insufficiencySevere semilunar valve insufficiencyTetralogy of Fallot with absent pulmonary valvesPulmonary atresia with severe tricuspid valve insufficiencyTruncus arteriosus with severe incompetent truncal valvesBilateral outflow tract obstructionIntracardiac tumorsSingle ventricle physiology with ventricular dysfunction

The majority of fetuses with CHD are well tolerated. This is because most cases have at least one well-functioning ventricle, contributing to redistribution of the systemic and pulmonary venous blood flow through the normal ventricle which can maintain the equivalent of a combined cardiac output without an increase in central venous and atrial filling pressures. Gembruch et al. [140] demonstrated that venous Doppler in fetuses with isolated CHD did not present sufficient alterations to be a reliable marker for screening purposes for CHD in second- and third-trimester fetuses. Abnormal venous Doppler results were mainly attributable to myocardial dysfunction and also to severe right heart obstruction even in the absence of congestive heart failure. Therefore, venous Doppler studies are clinically helpful in indirectly monitoring cardiac function in fetuses with CHD. When abnormal central venous flow patterns do occur in association with cardiac defects in utero, they are usually caused by other processes occurring simultaneously that affect ventricular compliance (e.g., endocardial fibroelastosis) or rhythm-related hemodynamics (e.g., complete heart block) [141].

CHD with dysfunction of both ventricles or with biventricular inflow or outflow obstruction, venous and atrial pressure load can occur and result in FHF. CHD associated with markedly increased afterload such as critical aortic stenosis, acute constriction of ductus arteriosus, truncus arteriosus with truncal valve stenosis are usually poorly tolerated, likely leading to abnormal ventricular dysfunction or FHF. CHD with volume load as commonly seen in atrioventricular and semilunar valve insufficiency are also likely to evolve FHF. For example, the Ebstein anomaly is usually related to severe tricuspid dysfunction and volume load (Figure 10), which eventually compromises left ventricular filling as well as global function and combined cardiac output, leading to hydrops fetalis [142]. Likewise, Inamura et al. [143] demonstrated that global left ventricular performance as assessed by the Tei index was abnormal in severe tricuspid valve dysplasia and likely to develop hydrops or result in mortality. Probably, markedly enlarged right-sided heart causes a mechanical compression of the left ventricle, leading to limitation of left ventricle filling. Additionally, in case of no antegrade flow through the right outflow tract and pulmonary trunk, total venous preload is redistributed to the left heart, resulting in left ventricular dysfunction secondary to excessive volume load and inability to maintain the equivalent of a combined cardiac output, ultimately evolving to FHF.

#### 5.7.3. External Cardiac Compression

External cardiac compression of the fetal heart can occur in cases of congenital diaphragmatic hernia (CDH) [144], congenital cystic adenomatoid malformation (Figure 11) [145], tumors such as mediastinal or pericardial teratoma [146,147] and massive pleural effusions [148]. These conditions can compress the heart or distort and compress the systemic veins returning to the heart, leading to reduced cardiac size and ventricular filling, resulting in low cardiac output and increased central venous pressures and eventually FHF or hydrops fetalis. Such lesions in some cases can also more directly influence lymphatic drainage and may be a primary cause of evolving fetal hydrops fetalis [149]. Fetal echocardiography usually shows diastolic dysfunction (increased Doppler E/A ratio and abnormal venous Doppler) together with reduced cardiac size (cardio-thoracic area ratio).

## 6. Conclusions

This review provides updates on FHF in terms of the pathophysiology of the common causes and practical points in prenatal diagnosis, mainly focusing on available techniques commonly used in actual practice. Though FHF due to any causes is typically associated with poor cardiac function, inadequate cardiac output to provide tissue perfusion and increased venous pressure as well as hydrops fetalis, the natural course of FHF development is different among different etiologies, as follows:

*Fetal anemia* causes an increase in cardiac output to provide adequate tissue perfusion. In spite of hypervolemia and increased cardiac output, the heart copes well with the longstanding workload because of a decrease in afterload due to anemic hypoxia together with low viscosity. Diastolic function seems less interfered with in the early stage, while for systolic function, ICT seems to be prolonged earlier than IRT. MCA-PSV is very helpful in early detection. Hydrops occurs much earlier than poor cardiac function and does not represent heart failure in most cases.

*High-output state caused by vessel shunting*, such as SCT or acardiac twin, systolic and diastolic dysfunction, is associated with both volume load and some degree of anemia; however,, the fetuses can increase compensatory production of red blood cells. Anemia is not usually severe and volume load becomes the main component of cardiac overload. Hydrops can occur when cardiac overload occurs, and cardiac function is gradually deteriorated.

In *TTTS*, the recipient has volume and pressure load without anemia, causing an increase in afterload due to vasoconstrictor release in response to hypervolemia, which leads to the development of poor cardiac function and hydrops. Diagnosis of TTTS is very helpful for early detection and specific treatment.

In *FGR*, increased afterload due to placental insufficiency results in blood flow redistribution, increased UA-PI and decreased MCA-PI. In the late phase, pressure load becomes severe and causes poor cardiac function with increased preload (AEDF/REDF in DV or UV-pulsations). UA-Doppler is very helpful in early detection.

In poorly controlled *diabetes mellitus*, as well as intrinsic myocardial disease, while preload and afterload are not obvious in early adaptation, the requirement of oxygen perfusion is markedly increased because of a high metabolic rate associated with hyperinsulinemia; ventricular hypertrophy and its consequences cause fetal heart failure and unexpected death without predictable high afterload or placental dysfunction as seen in FGR.

*Low output state* caused by an increase in afterload is associated with outflow tract obstruction. Heart failure is first developed on the affected side, such as in increased pulmonary pressure; rather than IVC or DV pressure being noted in a hypoplastic left heart or critical aortic stenosis, while umbilical artery waveforms seem to be preserved.

Understanding the pathophysiology and clinical courses of various etiologies of FHF can help physicians make prenatal diagnoses and serve as a guide for counseling, surveillance and management.

## Figures and Tables

**Figure 1 diagnostics-13-00779-f001:**
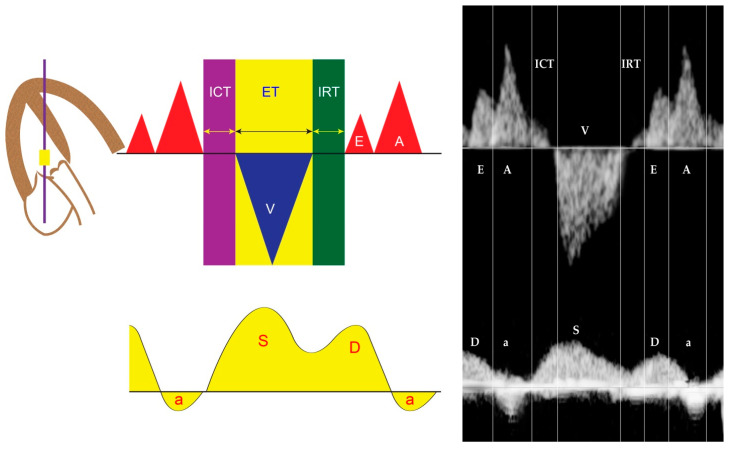
Cardiac cycle and Tei index of the left ventricle (placing the sampling cursor just below the mitral valve and outflow): correlation between Doppler waveforms of intra-cardiac flow and venous flow in the inferior vena cava (a: A-wave; D: D-wave; ET: ejection time; ICT: isovolumetric contraction time; IRT: isovolumetric relaxation time; S: S-wave; V: ventricular contraction).

**Figure 2 diagnostics-13-00779-f002:**
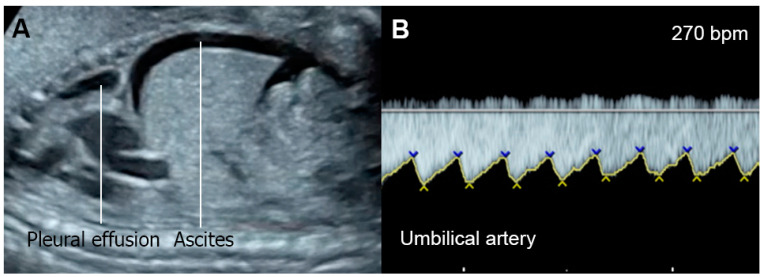
Supraventricular tachycardia: (**A**) hydropic signs; (**B**) high end-diastole velocity due to rapid heart rate; (**C**) ductus venosus Doppler: reversed A-wave, no D-wave due to rapid heart rate; (**D**) venous pulsations in the umbilical vein.

**Figure 3 diagnostics-13-00779-f003:**
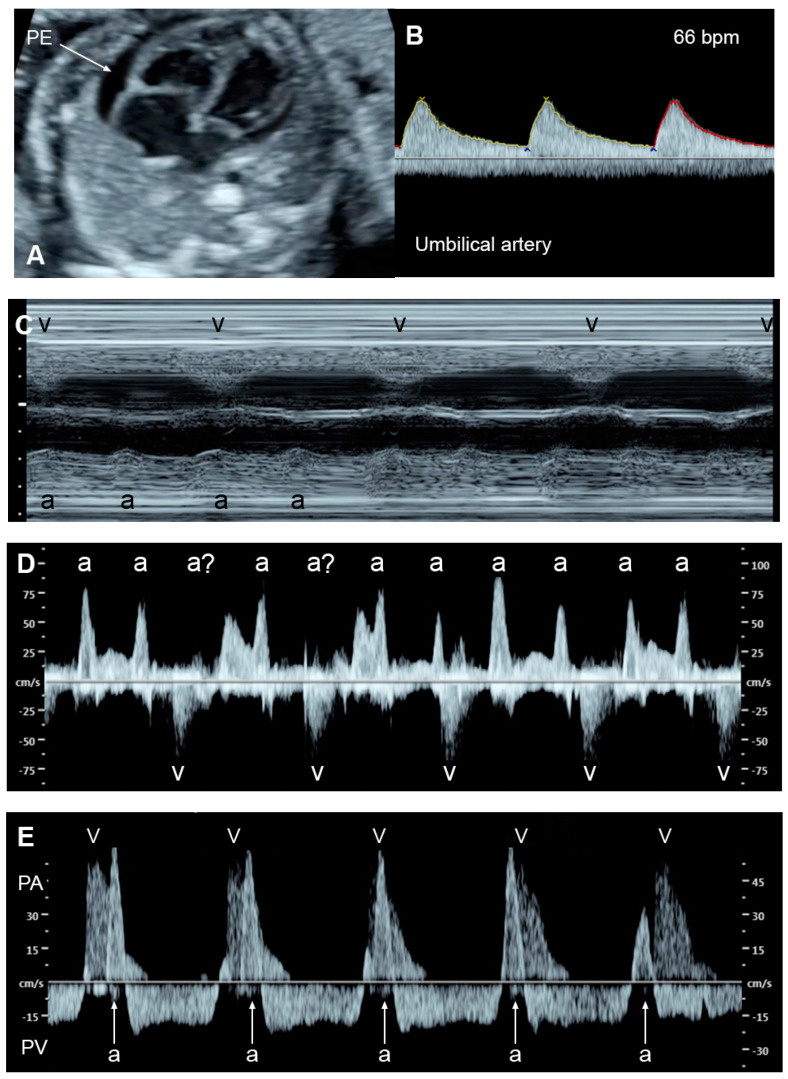
Complete heart block: (**A**) slightly enlarged heart with pericardial effusion (PE); (**B**) Ven-tricular rate of 66 bpm in the umbilical artery; (**C**–**E**) dissociations between atrial and ventricular contraction with ventricular bradycardia in M-mode (**C**) cardiac Doppler (**D**), and PW-Doppler of the pulmonary artery (PA) and pulmonary vein (PV) (**E**).

**Figure 4 diagnostics-13-00779-f004:**
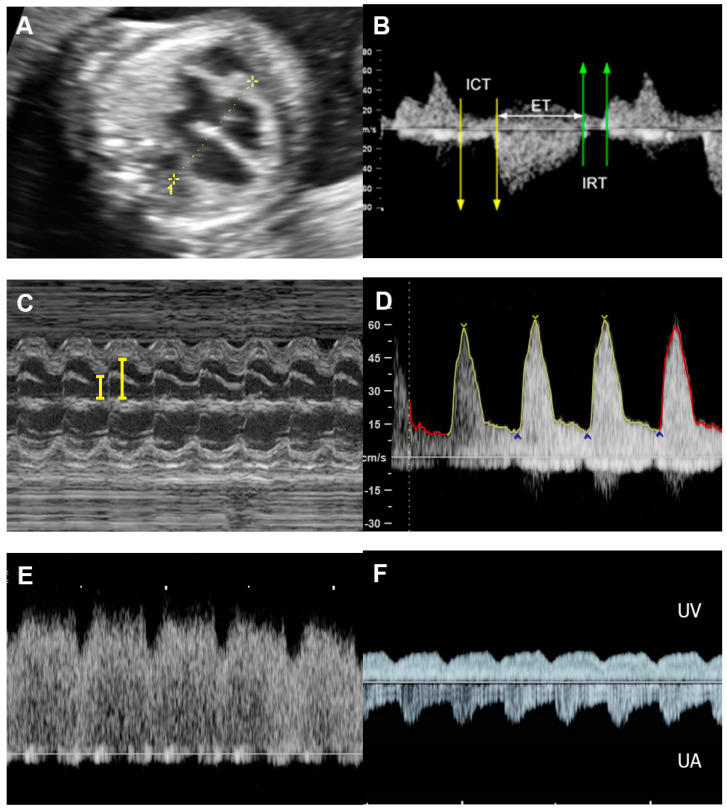
Hydrops without heart failure in Hb Bart’s disease: (**A**) cardiomegaly with effusions; (**B**) slightly increased Tei index (0.49) with prolonged ICT (42 ms); (**C**) normal shortening fraction 38%; (**D**) High MCA-PSV (60 cm/s); (**E**) markedly increased forward A-wave of the ductus venosus; (**F**) normal Doppler waveforms in the umbilical artery (UA) but pulsations in the umbilical vein (UV).

**Figure 5 diagnostics-13-00779-f005:**
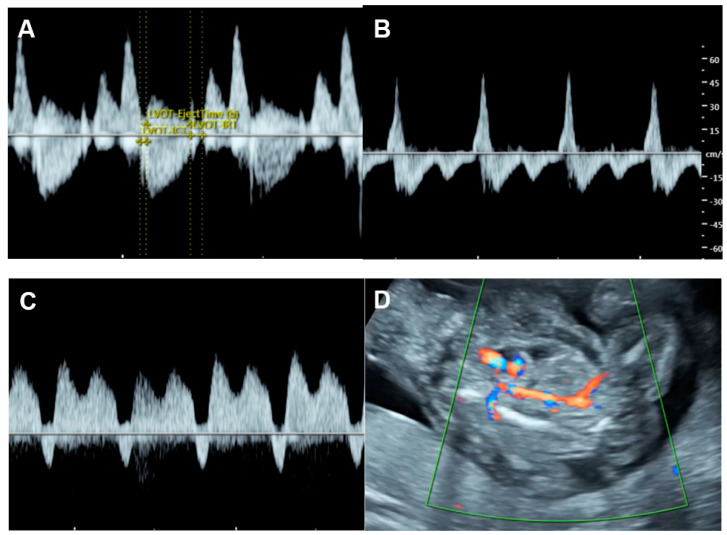
Cardiac dysfunction of the pumping twin in twin reversed arterial perfusion sequence: (**A**) increased Tei index (0.61) with prolonged IRT (46 ms); (**B**) Doppler flow in IVC; markedly re-versed A-wave (45 cm/s); (**C**) reversed A-wave in the ductus venosus; (**D**) acardiac twin with vas-cularization.

**Figure 6 diagnostics-13-00779-f006:**
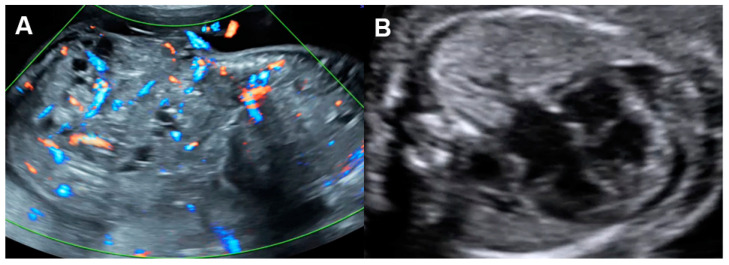
Cardiac dysfunction of a fetus with sacrococcygeal teratoma (SCT): (**A**) vascularized SCT; (**B**) global cardiomegaly with pericardial effusion; (**C**) increased Tei index (0.58) with prolonged IRT (44 ms); (**D**) Doppler markedly reversed A-wave in the ductus venosus.

**Figure 7 diagnostics-13-00779-f007:**
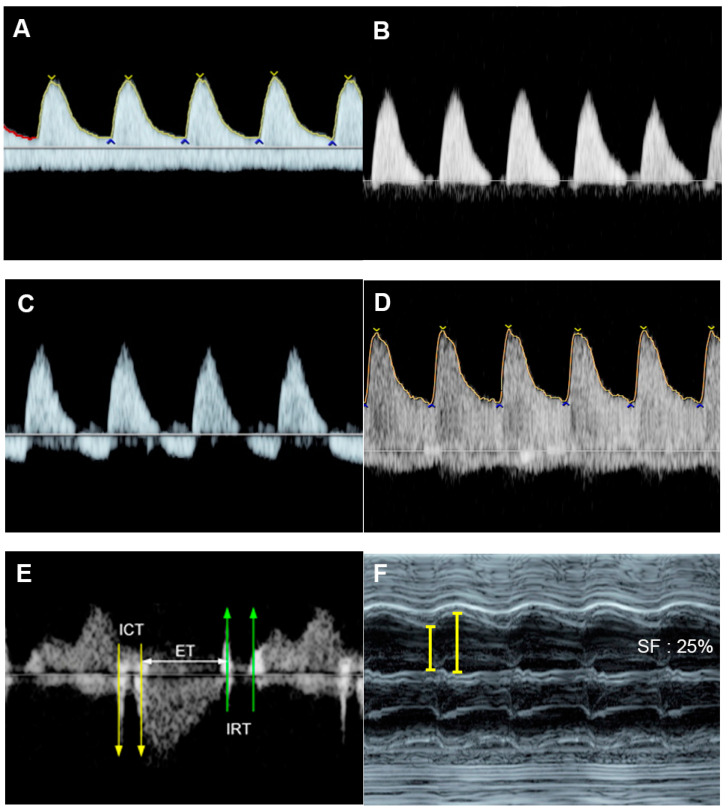
Fetal growth restriction: (**A**–**C**) Progressive changes of Doppler waveforms of the umbilical artery from low-end-diastolic flow at 27 weeks, to absent end-diastolic flow (**B**), and reversed end-diastolic flow (**C**) at 32 weeks; (**D**) Increased end-diastolic flow in the middle-cerebral artery (cephalization); (**E**) increased Tei index (0.61) with prolonged IRT (0.52 ms); (**F**) shortening fraction of 25%. (**G**) reversed A-wave in the ductus arteriosus; (**H**) pulsations in the umbilical vein at 32 weeks.

**Figure 8 diagnostics-13-00779-f008:**
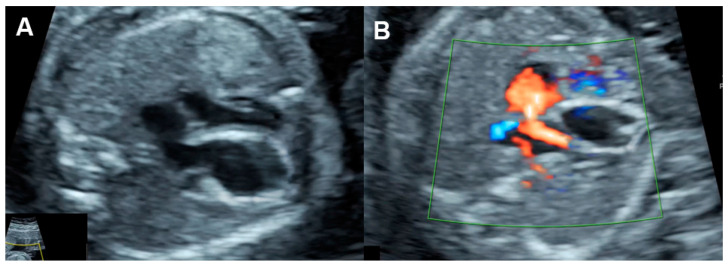
Critical aortic stenosis: (**A**,**B**) four-chamber views show enlarged LV with echogenic endocardium (fibroelastosis), mitral regurgitation, reversed flow across the foramen ovale; (**C**) markedly increased reversed A-wave in the pulmonary vein, indicating high pressure; (**D**) M-mode shows markedly poor contractility of the LV, compared to RV.

**Figure 9 diagnostics-13-00779-f009:**
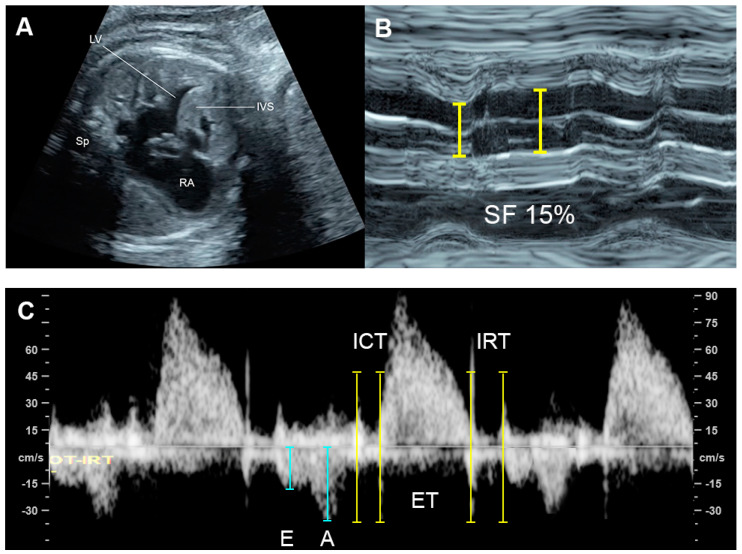
Hypertrophic cardiomyopathy: (**A**) four-chamber view: thickened ventricular wall and interventricular septum; (**B**) M-mode: poor contractility with a shortening fraction of 15%; (**C**) Car-diac PW Doppler: Low E/A ratio, prolonged IRT and increased Tei index (ICT: 40 ms; ET 154 ms; IRT 54 ms).

**Figure 10 diagnostics-13-00779-f010:**
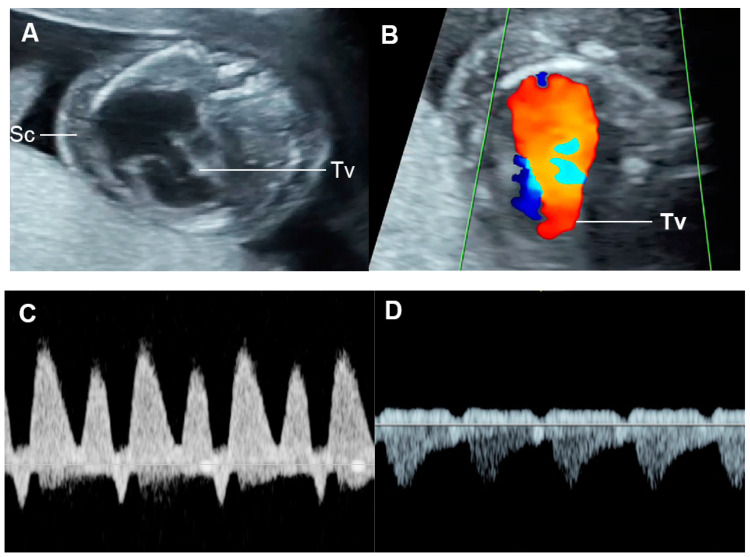
Ebstein anomaly with hydrops fetalis: (**A**) four-chamber view: cardiomegaly, low inser-tion of septal leaflet of the tricuspid valve (Tv) and subcutaneous edema (Sc); (**B**) color flow mapping of the four-chamber view: severe tricuspid regurgitation with originating point near the apex; (**C**) ductus venosus Doppler: reversed A-wave; (**D**) venous pulsations of the umbilical vein.

**Figure 11 diagnostics-13-00779-f011:**
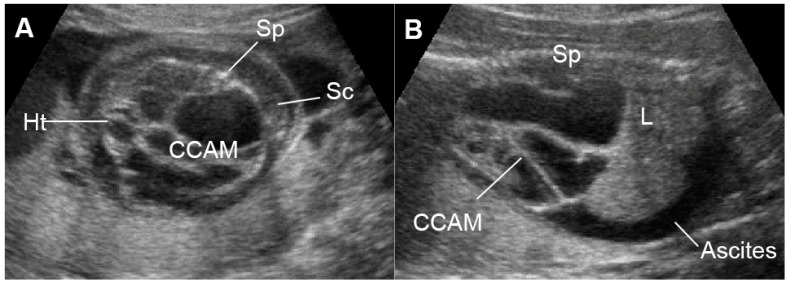
Large cystic adenomatoid malformation (CCAM) type I with hydrops fetalis: (**A**) cross-sectional scan at the four-chamber view: small heart compressed by the cyst and marked subcutaneous edema (Sc); (**B**) sagittal scan of the fetal trunk: CCAM occupying the entire chest and ascites (L: liver; Sp: spine).

**Table 1 diagnostics-13-00779-t001:** Parameters for evaluation of fetal cardiac function.

Parameters	Abnormal Changes	Interpretation
**Dimension**		
Cardiothoracic area ratio	Increased >35%	Cardiac enlargement
Cardiothoracic diameter ratio	Increased >95th centile	Cardiac enlargement
**Inflow characteristics**		
Filling time fraction	Decreased	Diastolic dysfunction
E/A ratio	Monophasic	Diastolic dysfunction
	Decreased	Diastolic dysfunction
	Increased	Volume loading/External compression
**Venous PW Doppler**		
Inferior vena cava	Reversed A-wave >20 cm/s	Diastolic dysfunction /increased venous pressure
	Decreased S-wave	Tricuspid regurgitation
Ductus venosus	Absent or reversed A-wave	Diastolic dysfunction/increased venous pressure
**Performance**		
Shortening fraction	Decreased (<28%)	Systolic dysfunction
	Increased	Reduced afterload/increased contractility
Ejection fraction	Decreased <50%)	Systolic dysfunction
	Increased	Reduced afterload/increased contractility
Cardiac output (Stroke volume)	Decreased (Z < −2)	Systolic dysfunction/poor filling
	Increased (Z > +2)	Reduced afterload/volume load
Tei index	Increased > 0.50	Global cardiac dysfunction
ICT: 28 (22–33) ms	Prolonged	Systolic dysfunction
IRT: 34 (26-41) ms	Prolonged	Diastolic dysfunction
Systolic strain/Strain rate	Increased	Reduced afterload
	Decreased	Reduced contractility
E/Vp (Color M-mode)	Increased	Diastolic dysfunction

(E/A = early/atrial ventricular filling; E/Vp = early/velocity of propagation; ICT = isovolumic contraction time; IRT = isovolumic relaxation time).

**Table 2 diagnostics-13-00779-t002:** Criteria for cardiovascular profile score.

	Normal	−1 Point	−2 Points
Hydropic signs	Absence of effusion	Abdominal **or** pleural, **or** pericardial effusion	Skin edema
Venous Doppler (umbilical vein: UV & ductus venosus: DV	Normal Doppler UV 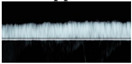 DV 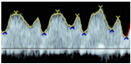	Reversed ductus venosus flow UV 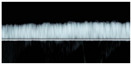 DV 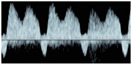	Pulsatile flow in the umbilical vein UV 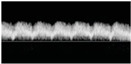
Heart size (Cardio-thoracic ratio)	≤35%	35–50%	>50% or <20%
Cardiac function	Normal function	Holosystolic TR, or ventricular shortening fraction < 28%	Holosystolic MR or TR dP/dt < 400, or monophasic inflow
Arterial Doppler (umbilical artery)	Normal Doppler 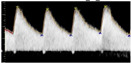	Absent end-diastolic flow 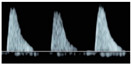	Reversed end-diastolic flow 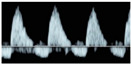

**Modified from** Huhta JC. Guidelines for the evaluation of heart failure in the fetus with or without hydrops. Pediatr Cardiol 2004;25:274-86. (DV: ductus venosus; TR: tricuspid valve regurgitation; UV: umbilical vein).

## Data Availability

The data of this report are available from the corresponding authors upon request.

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
