# Peer review of "Prenatal Diagnosis of Fetal Heart Failure"

_diagnostics, 2023, doi:10.3390/diagnostics13040779_

Round 1
Reviewer 1 Report (New Reviewer)
The review is hard to read.
I suggest a rearrangement of informations.
The abstract requests a rewriting. In addition, the introduction is a copy with minimum insertions of the abstract.
The pathophysiology is not very clear and in addition is based at only 4 bibliographic informations. For a period of more 30 years is impossible to find only 4 papers regarding this sujet.
In the section 4 ”Prenatal assessment of fetal hemodynamics” I suggest to use an special editor to write the differents formulas. Some paragraphs of this section are based on a small number of bibligraphic indications - requests supplementation.
Section 5 ”Common causes of FHF” has a title that must be changed. I propose: ”Clinic and echographic features of different forms of FHF”. In the list presented in the begining of this section are showed seven pathologies, but in the text are detailed only five.
The paper contains many figures, but the source of this images is not precised. Please, indicate the source of images.
Author Response
Reviewer 1(response in red in the revised MS)
Comments and Suggestions for Authors
The review is hard to read.
I suggest a rearrangement of informations.
The abstract requests a rewriting. In addition, the introduction is a copy with minimum insertions of the abstract.
Response: In revised MS, the abstract is rewritten and no a copy of abstract inserting to the introduction.
The pathophysiology is not very clear and in addition is based at only 4 bibliographic informations. For a period of more 30 years is impossible to find only 4 papers regarding this suject.
Response: In revised MS, pathophysiology is more elaborated with adding the citations from 4 to 20.
In the section 4 ”Prenatal assessment of fetal hemodynamics” I suggest to use an special editor to write the differents formulas. Some paragraphs of this section are based on a small number of bibligraphic indications - requests supplementation.
Response: Since the formula is simple, we make a request to not use a special editor to create the formula. In the section 4, all paragraph are now properly added the bibliographic citations, as highlighted.
Section 5 ”Common causes of FHF” has a title that must be changed. I propose: ”Clinic and echographic features of different forms of FHF”. In the list presented in the beginning of this section are showed seven pathologies, but in the text are detailed only five.
Response: The title is changed as suggested. In revised MS, the item 6 (Congenital heart defect) and 7 (External cardiac compression) are included, as highlighted at the end of MS before the section “Conclusion”.
The paper contains many figures, but the source of this images is not precised. Please, indicate the source of images.
Response: All figures in this review belong to us. All are original from our experience, collected aimed to serve as educational tool.
Reviewer 2 Report (New Reviewer)
Authors describe a very interesting and well written manuscript analyzing the clinical performance of a cardiovascular profile score (CPS) in different fetal diseases associated with fetal heart failrues.
The ms is well structured in each section with high-quality scheme and prenatal imaging.
However, authors must spell oput every single acronym e.g. "E", "A", "FAM", "FAM-MAPSE" and use the same aconym. For example, in Table 1 authors use "IVCT" while only after a while in Figure 1 they use "ICT". This fact generates confusion and must be avoided! Use the same symbol, spell out every acronym!
Again, in Figure 1 authors report in the Legend "...venous flow". It must be specify which type of venous flow...I think they have sampled the inferior vena cava (IVC). If so, please add correctly which venous vessels was studied as well as the need to specify in the Legend each acronym used. What is a, S and D vaves...?
Author Response
Reviewer 2 (response in blue in revised MS)
Comments and Suggestions for Authors
Authors describe a very interesting and well written manuscript analyzing the clinical performance of a cardiovascular profile score (CPS) in different fetal diseases associated with fetal heart failrues.
The ms is well structured in each section with high-quality scheme and prenatal imaging.
However, authors must spell oput every single acronym e.g. "E", "A", "FAM", "FAM-MAPSE" and use the same aconym. For example, in Table 1 authors use "IVCT" while only after a while in Figure 1 they use "ICT". This fact generates confusion and must be avoided! Use the same symbol, spell out every acronym!
Response: In revised MS, all the abbreviations are spelled out at the first time, and uniformly, as suggested.
Again, in Figure 1 authors report in the Legend "...venous flow". It must be specify which type of venous flow...I think they have sampled the inferior vena cava (IVC). If so, please add correctly which venous vessels was studied as well as the need to specify in the Legend each acronym used. What is a, S and D vaves...?
Response: In revised MS, the legend of Fig 1 is modified as suggested.
This manuscript is a resubmission of an earlier submission. The following is a list of the peer review reports and author responses from that submission.
Round 1
Reviewer 1 Report
It is a very descriptive manuscript about fetal pathologies leading to fetal heart failure. I appreciate the figures.
This is destinated to the section Pathology and Molecular Diagnosis of Diagnostics journal , to prenatal diagnosis .
“this review focuses on a straightforward method for rapid evaluation of the fetus with possible FHF, based on common echocardiographic examinations in prenatal detection of fetal heart failure and common disorders associated with cardiac dysfunction, culminating in heart failure, together with their it is not very clear about what method is this article?
There are many mistakes: a normal fetus maybe refer to a healthy fetus, parvovirus B19 refers to parvovirus B19 infection ; fetal anemia can be a manifestation of TAPS ? etc
Evolution of fetal heart failure: I suggest to explain the mechanisms of fetal heart failure and the progression generally also the compensatory mechanisms.
3. Prenatal assessment of FHF : “Clinically, FHF, regardless of the underlying causes, is usually encountered unexpectedly in most cases at the time of routine fetal ultrasound examinations of asymptomatic mothers.” It is not true because worldwide there is a screening for fetal malformations and firstly we diagnose IUGR, cardiomyopathy, arrhythmia etc, and progression to FHF if not correctly managed.
For fetal anemia , TAPS – nowadays we treat or deliver the baby after 32 weeks of gestation.
“Several systemic disorders are associated with an increase in combined cardiac output such as arteriovenous malformations, a recipient of twin-to-twin transfusion syndrome or twin reversed arterial perfusion (TRAP) sequence” – these are not systemic disorders medically speaking
The message for this review is not clear and in the case of readers with expertise in prenatal diagnosis what is the message or utility? For people not doing ultrasound for prenatal diagnosis is too technique , too difficult to follow with no key messages.
Author Response
Reviewer 1
It is a very descriptive manuscript about fetal pathologies leading to fetal heart failure. I appreciate the figures. This is destinated to the section Pathology and Molecular Diagnosis of Diagnostics journal , to prenatal diagnosis .
“this review focuses on a straightforward method for rapid evaluation of the fetus with possible FHF, based on common echocardiographic examinations in prenatal detection of fetal heart failure and common disorders associated with cardiac dysfunction, culminating in heart failure, together with their it is not very clear about what method is this article?
Response: Concerning method of cardiac function, we make more clear, at page 3. (Since many modern sonographic methods for fetal cardiac function are currently introduced, we have reviewed and focused on those with practical use in actual practice. Also we provide insights on different pathophysiology and clinical findings on common different causes of fetal heart failure.) Note that, we focus on prenatal diagnosis in terms of clinical use in actual practice, relevant to “Diagnostics”.
Concerning method or review, in revised MS, we add under subheading “Methods of review” on page 2, as highlighted.
There are many mistakes: a normal fetus maybe refer to a healthy fetus, parvovirus B19 refers to parvovirus B19 infection ; fetal anemia can be a manifestation of TAPS ? etc
Response: In revised MS English is now edited by professional service. The certificate of English proof-editing is attached.
Evolution of fetal heart failure: I suggest to explain the mechanisms of fetal heart failure and the progression generally also the compensatory mechanisms.
Response: In revised MS, the mechanism of FHF is added and modified in heading “Evolution of fetal heart failure”, on page 3.
- Prenatal assessment of FHF : “Clinically, FHF, regardless of the underlying causes, is usually encountered unexpectedly in most cases at the time of routine fetal ultrasound examinations of asymptomatic mothers.” It is not true because worldwide there is a screening for fetal malformations and firstly we diagnose IUGR, cardiomyopathy, arrhythmia etc, and progression to FHF if not correctly managed.
For fetal anemia , TAPS – nowadays we treat or deliver the baby after 32 weeks of gestation.
“Several systemic disorders are associated with an increase in combined cardiac output such as arteriovenous malformations, a recipient of twin-to-twin transfusion syndrome or twin reversed arterial perfusion (TRAP) sequence” – these are not systemic disorders medically speaking
Response: In revised MS, the word “systemic” is deleted and English is now edited by professional service.
The message for this review is not clear and in the case of readers with expertise in prenatal diagnosis what is the message or utility? For people not doing ultrasound for prenatal diagnosis is too technique , too difficult to follow with no key messages.
Response: Key messages are added at the end of “Introduction”, and “Conclusion” is added at the end of MS,

Reviewer 2 Report
Dear authors,
Thanks for submitting this article that sought to review the causes of heart failure and its prenatal diagnosis. Although the article is well written, I have some ethical concerns regarding self-citations. Every author has the following self-citations:
Kasemsri Srisupundit: 12
Suchaya Luewan : 17
Theera Tongsong: 19
I consider this amount of citations unnecessary, and its presence is not justified at all.
The manuscript should undergo a process of proof reading in order to correct some punctuation mistakes such as brackets / verbs missing...
But the most important concern is that there are no Methods or Conclusion sections. I think those sections are very important, even more, in an article that tries to carry out a literature review. Furthermore, the inclusion or exclusion criteria of the articles should be introduced.
As a suggestion, a section that includes literature search results and risk of bias should be introduced.
There is no solid conclusion in this paper and the information reported in this article is merely academic and can be found in precedent publications. So, the work is not a significant contribution to the field.
Author Response
Reviewer 2
Comments and Suggestions for Authors
Thanks for submitting this article that sought to review the causes of heart failure and its prenatal diagnosis. Although the article is well written, I have some ethical concerns regarding self-citations. Every author has the following self-citations:
Kasemsri Srisupundit: 12; Suchaya Luewan: 17; Theera Tongsong: 19
I consider this amount of citations unnecessary, and its presence is not justified at all.
Response: In the revised MS, we reduce the number of self-citations (to be Srisupundit: 10; Luewan: 13; Tongsong: 13). Note that, since our group has extensively studied on fetal response to fetal anemia caused by Hb Bart’s disease, which is rarely studied by other group. Accordingly, we have a rather great number of self-citations on this aspect.
The manuscript should undergo a process of proof reading in order to correct some punctuation mistakes such as brackets / verbs missing...
Response: In the revised MS, English has been corrected and edited by professional English editing service, as the attached file of English editing certificate.
But the most important concern is that there are no Methods or Conclusion sections. I think those sections are very important, even more, in an article that tries to carry out a literature review. Furthermore, the inclusion or exclusion criteria of the articles should be introduced.
As a suggestion, a section that includes literature search results and risk of bias should be introduced.
Response: In the revised MS, the method section is added in “Introduction”, as highlighted in subheading “Methods”.
There is no solid conclusion in this paper and the information reported in this article is merely academic and can be found in precedent publications. So, the work is not a significant contribution to the field.
Response: In the revised MS, the conclusion section is added at the end of the MS, as highlighted in subheading “Conclusion”. Actually, we point out that, different from other previous review, we elaborate and conclude that high-output heart failure caused by anemia (based on our extensive studies on fetal anemia using Hb Bart’s disease as a study model) and volume overload in TTTS has different mechanism and prognosis, etc.

Round 2
Reviewer 1 Report
On page 3 there is a repeated paraghraph - Table 1
Reviewer 2 Report
Dear authors,
Thank you for taking the time to address comments on the manuscript entitled: " Prenatal Diagnosis of Fetal Heart Failure".
The manuscript has been much improved, but there are still some issues to clarify.
Although the self-citations have been reduced, there is still a huge amount of them. Although the group has widely studied anemia caused by Hb Bart´s disease, this MS is not focused on this aspect. In fact, the title of the article refers to Fetal heart failure and Hb Bart´s disease is one of the causes among others.
The Methods section should be placed in a different section and not in the Introduction, and furthermore, there is no information about the inclusion or exclusion criteria and a flow chart of the articles initially selected and the final amount of them after the bibliographic revision. So, this section must be improved.
Although this article is introduced as a narrative review, following the SANRA (Scale of the Assessment of Narrative Review articles- I attach the article), the quality of this narrative review based on formal criteria is low:
-There is no justification of the article´s importance for the readership
-No aims of questions are formulated
-The search strategy is not acceptable
-Key statements are supported by references (but a lot of self-citations)
-Appropriate evidence is introduced selectively.
-Data are often not presented in the most appropriate way.
I still think that this paper and the information reported in this article is merely academic and can be found in precedent publications.
